# The role of tourism in service sector employment: Do market capital, financial development and trade also play a role?

**Avishek Khanal** *, **Mohammad Mafizur Rahman**, **Rasheda Khanam**, **Eswaran Velayutham**

School of Business, Faculty of Business, Education, Law and Arts, University of Southern Queensland, Toowoomba, Australia

* avishek_khanal@yahoo.com, avishek.khanal@usq.edu.au

## Abstract

Workers' living standards have recently deteriorated in the service sector throughout the world, although a few decades ago, service was among the fastest growing sectors in industrialised nations. However, in recent years, in service sectors tourism especially has been drying up. This paper examines the symmetric and asymmetric effects of tourism, market capital, financial development, and trade on service sector employment in Australia from the period 1991–2019. The results of the cointegration tests, notably the ARDL and NARDL bound tests, reveal that the variables are related in the long run. The positive effect of tourist arrival on service sector employment in Australia is confirmed by long-run estimates from both ARDL and NARDL approaches. Similarly, both approaches also confirm the long-run positive relation of financial development. However, while ARDL shows long-run negative and positive associations of market capital and trade, respectively, the opposite is found in the case of the NARDL approach. As a result, policy proposals like planning and initiating tools for ensuring consistent international arrivals and easing of entry requirements have been recommended by this study to assist Australia in enhancing service sector employment, thus promoting economic development.

## Introduction

The industrial revolution strengthened manufacturing units by easing and enhancing the volume of production, and thus the manufacturing sector is known as the engine of growth. However, over time, the contribution of manufacturing sectors globally has constantly been declining in terms of national income and employment, while the contribution of the service industry has been increasing. The service sector in an economy includes diverse industries and accounts for a substantial contribution to the country's growth and development. In developed countries, the service sector accounts for around 70% of the gross domestic product (GDP), while according to an earlier estimate, around 79% of Australia's economic activities were from the service sector [1]. Service sector employment in Australia was 78.4% in 2019, a decline from 85% in 2006, and an increase from 76% in 1985, indicating a fluctuation in

**Data Availability Statement:** All datas are available from the WDI world bank database (https://datacatalog.worldbank.org/dataset/world-development-indicators).

**Funding:** The authors received no specific funding for this work.

**Competing interests:** The authors have declared that no competing interests exist.

sectoral employment [1]. Australia's key services include education and tourism, recreational trade, FinTech and environmental services, while the most significant service exports are professional services, and business travel services [1].

Although workers' living standards deteriorated in a service economy, a few decades ago service was among the fastest growing sectors in industrialised nations [2, 3]. The argument Fuchs [4] made long back that economic development contributed to the rise of service employment seems valid today since higher family income influences more spending on various services.

Tourism dynamics are not confined to a particular area. Although Europe attracted many visitors in 2014, the Asia Pacific and Africa have enjoyed the highest growth during the decade that ended 2014 [5]. International tourism directly impacts both the economy and employment [5, 6]. Tourism has attracted researchers to find its association with numerous factors and determinants, including emission and environment [7], energy consumption [8], foreign exchange rate [9], and economic development [10]. The tourism industry is predominantly resource-based and is influenced by the climate and landscape of the destination, heritage, and cuisine. Performance of the labour-intensive industries, including hotels, transports, and restaurants, amplifies the potential travellers' motivation [6]. The hotel, transport, and other facilities use energy consumption which impacts the environment [11, 12].

Despite such advancement in research, evidence about the service industry, employment, and related factors is inadequate, particularly from the Australian perspective, while this sector can potentially influence growth [13–15]. Against this backdrop, this research aims to investigate the influences of tourist arrival, market capital, financial development, and volume of trade on service sector employment in Australia. The research findings are likely to contribute to the existing body of literature in two ways: first, by compiling seemingly different determinants in the analysis of employment in the service sector, as this research acknowledges the importance of various service-oriented industries. Second, using an established dataset to recognise the impact of multiple service industries on service sector employment as this is not readily available in the existing literature.

## Background and literature review

Sectoral shifts in employment over time are viewed as structural transformation [16], and this field of research has gained attraction. For instance, in the Chinese context, Wang and Zhang [17] found that better transportation infrastructure results in employment density in the service sectors. They also found that road transportation promoted service sector employment more than railway and inland water transport. Walmsley, Koens and Milano [18] have found that overtourism, excessive numbers of tourists in a particular tourist spot, has the potential to impact wage and to divide labor market. In the Indian context, R&D, legal, media and broadcasting services have all been identified as potential sectors for future growth [19]. Education, health, growth in population, and inflation positively influence service sector employment, while the impact of political conflict is the reverse, as found in Nepal [20]. Nonetheless, the impact of various service-oriented industries on this sectoral employment is rarely investigated.

The travel and tourism industry plays a vital role in the global economy. In 2010, more than 235 million employees were linked to this sector. Later, although it was with a minor decline in 2008–09, it grew consistently until the end of 2019, when COVID impacted the whole world. According to International Labour Organisation (ILO), tourism was expected to grow around nine per cent of total GDP [21] but tourism has been one of the sectors that is most harshly impacted from the spread of COVID-19. During the harshest impact of COVID-19, many

sectors including tourism were subject to restrictions and anti-pandemic measures due to their higher potential to spread COVID-19 [22]. However, during post-COVID investigation, Scarlett [23] found that tourism has a significant positive impact on economic growth. The author has also found that a relative increase in tourism to GDP is likely to positively impact FDI inflow. In Australia, the Australian Bureau of Statistics (ABS) confirmed that since September 2020, only three months (from May to July 2021) have welcomed more than 60 thousand overseas arrivals, while some months have seen 20–40 thousand arrivals [24]. Tourism-dependent nations, particularly the small island developing states (SIDS), experience a higher proportion of employment in the service industry. In the case of a decline in tourism, substantial migrated labour and a small proportion of indigenous labour can be found redundant as was the case in Malaysia from 2007 to 2009 [6]. Hence, it is assumed that a boom or recession in tourism impacts employment in the sector, and an inferential statistical analysis is likely to provide a piece of evidence. Fig 1 shows the upward increasing trend of tourist arrivals and employment in services (% of total employment) (modeled ILO estimate).

Concerning the contribution or impact of the stock market on economic development, many investigations have been conducted across the world, for instance, in Belgium [26], in parts of Africa [27], and in many developing countries [28]. It has been found that a structured stock market in the long run results in the economic growth of a country [27], and stock market-based funding is a determinant of economic growth [26]. Irani, Katircioglu and Gokmenoglu [29] have found that the tourism stock price is impacted by foreign tourist arrivals and the price is more sensitive to changes in tourist arrival. Growth in the service sector as well eventually impacts the sustained growth of an economy [30]. However, the influence of the structured and the developed stock market on service sector employment in general and from the Australian perspective is an underexplored area of research.

Among many other indicators, the volume and quality of domestic credit to the private sector is also an indicator of an increase in investment and financial development [31–34]. Isaeva et al. [35] argue that higher receipts from tourism result in a higher share of domestic credit to private sector and vice versa. Moreover, well-structured financial systems enhance the tourism

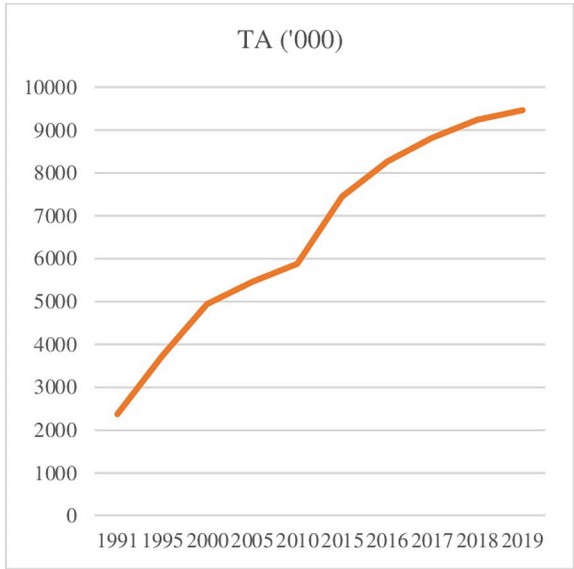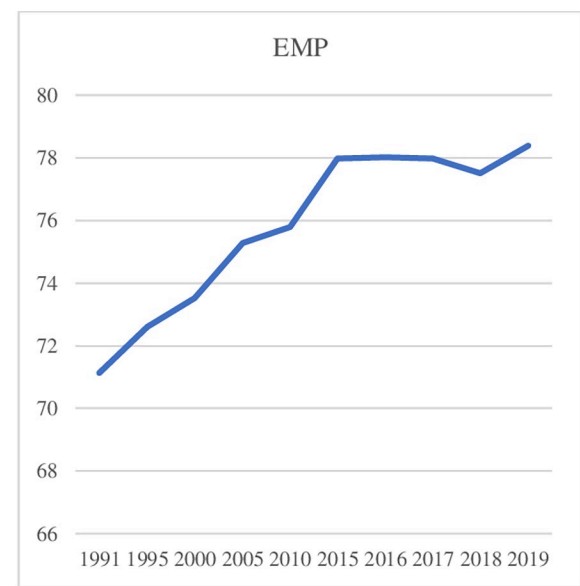

**Fig 1. Tourist arrivals and employment in service in Australia.** Source: World Bank [25].

development of a nation [35]. However, other studies have ascertained that domestic credit may not boost economic growth [36, 37]. Many researchers across the world have investigated the causal relationship between economic growth and domestic credit, for instance, China [38], Kenya [39], Tunisia [40], and Brazil, Russia, India, China, and South Africa (BRICS) [41]. In addition, domestic credit has also been investigated for trade [42]. Yousaf et al. [43] have found that board capital positively impacts the performance of tourism service providers. In the context of the Gulf region, tourism sectors are found to rely more on short-term than long-term debt [44]. However, a causal relationship regarding service sector employment, particularly focusing on Australia, is under explored.

The other variable considered for this research is trade, which generally results in employment and income in the global economy [45]. In 2005, more than 70% of the industrialised nations' employment was in a service sector consisting mostly of non-tradable activity [46]. Trade policies impact employment, labour market institutions, and policies. While trade and trade liberalisation may be the reason for company closures and thereby job losses in one part, start-ups or new firms start to commence operations in other parts of an economy requiring more labour [46, 47]. Hence, trade policies have attracted numerous researchers in the last couple of decades. However, most researchers concentrated on manufacturing employment [46], and provided sufficient reasons for further studies to focus on service sector employment. Trade, in general, results in employment and income in the global economy. Trade policies impact employment, labour market institutions and policies. While trade and trade liberalisation are the reason for company closures and thereby job losses in one part, start-ups or new firms start to operate in other parts of an economy requiring more labour [46, 47]. Ehigiamusoe [48] has identified that the individual causality between tourism and growth shows a bidirectional connection. It is also found that tourism is a significant predictor of financial development and economic growth [48].

In one of the earlier studies, Armah [49] showed varied trade-related employment gains with the proportion of women the minority of labour. A recent investigation identified that volume of service exports plays a pivotal role to optimise service sector employment, particularly in the case of China [50]. Trade openness and other variables positively affect sectoral shift towards service industries [51]. In the German context, service sector employment has grown while manufacturing jobs are declining; however, these were not attributed to the rising trade with other nations [52]. In the case of Cambodia, trade shock was found to impact different manufacturing industries differently, while there was no impact on the informal sector [53].

There are at least two reasons identified by Kelle and Kleinert [54] for the growing importance of service trade: service is becoming increasingly vital in modern economies, and technological advancement has made service increasingly tradable. In the case of Australia, a limited volume of trade involves the advanced producer services sector, which is well proportioned [55]. The absence of suitable host conditions is a reason for such low volume. The services sector in Australia employs 80% of Australians and accounts for more than 70% of GDP. According to Australian Government's department of foreign affairs and trade (DFAT), the involvement of international service trade with countries resulted in around 22% of total exports in 2016 [56].

Dignity in the tourism industry is another field that has attracted numerous researchers. Tourism employment has resulted in violation of dignity of indigenous people [57]. Tourism and sustainability nexus has become another dimension in the recent literature, while it has been argued that research should be carried out on the integration of work and workers focusing dignity [58]. Tosun et al. [59] argued that in the developing world considering tourism as a developmental instrument and thereby benefitting from it would be difficult. However, in the perspective of develop world, the condition may be different.

## Data and methodology

### Data

The study examined the association of international tourism in Australia and service sector employment. In addition, to avoid omitted variable bias, some additional explanatory variables i.e., market capital (listed domestic companies in current US$), financial development (domestic credit to the private sector as a percentage of GDP) and trade (percentage of GDP) were also incorporated. The data extracted in this paper was created from time series observation in Australia from 1991 to 2019. All the data were collected from World Development Indicators (WDI) [25]. The data source and variables description are presented in Table 1.

### Empirical model

The following model has been developed to assess the effect of tourist arrival, market capital, and financial development on service sector employment in Australia following Rahman, Shahbaz and Farooq [45].

$$\text{LN } EMP_t = \beta_0 + \beta_1 \text{ LN } TA_t + \beta_2 \text{ LN } MC_t + \beta_3 \text{ LN } FD_t + \beta_4 \text{ LN } TR_t + \varepsilon_t \quad (1)$$

Where $\varepsilon_t$ is the error term, while LN $EMP_t$, LN $TA_t$, LN $MC_t$, and LN $FD_t$ are the natural logarithms of employment in the service sector, tourist arrivals, market capital and financial development, respectively. This study used the logarithmic forms of the variables of interest to stabilise the variance of the series [60].

### Unit root test

The stationarity of time-series data is important since the results of causality tests rely on it, and macroeconomic variables frequently have a unit root. If the first and second moments of a stochastic process are time-invariant, the process is said to be stationary in which statistical qualities do not change [61].

If a variable's first difference is stationary, it is integrated of order I(1) [62, 63]. The auxiliary equation below was taken from Lütkepohl, Krätzig and Phillips [61].

$$\Delta y_t = \mu + \propto_i y_{t-1} + \sum_{i=1}^{k} \pi_i \Delta y_{t-i} + \varepsilon_t \quad (2)$$

**Table 1. Variable's description.**

| Variable | Description | Definition | Source |
|---|---|---|---|
| **Employment (EMP)** | Employment in services (% of total employment) | Employment is defined as persons of working age who were engaged in any activity to produce goods or provide services for pay or profit, whether at work during the reference period or not at work due to temporary absence from a job, or to working-time arrangement. The services sector consists of wholesale and retail trade and restaurants and hotels; transport, storage, and communications; financing, insurance, real estate, and business services; and community, social, and personal service | WDI |
| **Tourist Arrivals (TA)** | Number of international tourist arrivals | International tourist arrivals; short-term visitors arriving | WDI |
| **Market Capital (MC)** | Market capitalisation of listed domestic companies (current US$) | Market capitalisation (also known as market value) is the share price times the number of shares outstanding (including their several classes) for listed domestic companies. Investment funds, unit trusts, and companies whose only business goal is to hold shares of other listed companies are excluded. Data are end of year values. | WDI |
| **Financial Development (FD)** | Domestic credit to the private sector (% of GDP) | Domestic credit to the private sector refers to financial resources provided to the private sector by financial corporations. | WDI |
| **Trade (TR)** | Trade (% of GDP) | Trade is the sum of exports and imports of goods and services measured as a share of gross domestic product. | WDI |

Where *it* denotes the relevant time-series variable, t denotes a linear deterministic trend, $\Delta$ is the first difference operator, $\propto_i$ denotes the parameter of interest, *k* denotes the maximum lag order, and $\varepsilon_t$ denotes the error term. If $|\propto i| < 1$, the series is trend stationary; conversely, when $|\propto i| \geq 1$, the series has the unit root and is thus not stationary. For more information on the time-series unit root test, see here. For further information on the time-series unit root test, see Hamilton [64] and Lütkepohl, Krätzig and Phillips [61].

## BDS test

The BDS test compares the null hypothesis that the data is distributed independently and identically (iid) to an unspecified alternative [65]. Its null hypothesis is that data in a time series is linearly dependent. The test is unique because it can determine non-linearities without being influenced by linear data dependencies. The BDS test is a two-tailed test, and the null hypothesis is rejected if the BDS test statistic is greater than or less than the critical values (e.g. if a = 0.05, the critical value = ±1.96).

## Cointegration analysis

This study utilised the Pesaran cointegration test, namely the autoregressive distributed lag (ARDL) bound test, to explore the long run and short run association among the dependent and explanatory variables, [66]. However, because of the possible asymmetric association among the variables, this study also used a nonlinear autoregressive distributed lag (NARDL) model developed by Shin, Yu and Greenwood-Nimmo [67].

**The symmetric analysis: ARDL bound testing technique.** The paper also uses the cointegration technique developed by Pesaran, Shin and Smith [66], particularly the ARDL bound test. The ARDL approach has gained popularity among scientists due to its benefits compared to other standard cointegration methods for identifying the symmetric association of service sector employment and other explanatory factors [68–70].

The bound test provides two asymptotic critical values when the independent variables are I (0) or I (1). If the F-statistic value is greater than the upper critical bound, I (1), it can be concluded that the variables are cointegrated and that there is a long-run relationship among them. The empirical expression of the ARDL bound test for cointegration is presented as follows:

$$\Delta \text{LNEMP}_t = \beta_0 + \beta_1 \text{LN}EMP_{t-1} + \beta_2 \text{ LN}TA_{t-1} + \beta_3 \text{ LN}MC_{t-1} + \beta_4 \text{ LN}FD_{t-1} +$$
$$\beta_5 \text{ LN}TR_{t-1} + \sum_{i=1}^{p}\alpha_1 \text{ }\Delta\text{LNEMP}_{t-i} + \sum_{i=1}^{p}\alpha_2 \text{ }\Delta\text{LNTA}_{t-i} + \sum_{i=1}^{p}\alpha_3 \text{ }\Delta\text{LNMC}_{t-i} + \qquad (3)$$
$$\sum_{i=1}^{p}\alpha_4 \text{ }\Delta\text{LNFD}_{t-i} + \sum_{i=1}^{p}\alpha_5 \text{ }\Delta\text{LNTR}_{t-i} + \varepsilon_t$$

Where *p* is the lag length, $\beta_0$ is constant and $\varepsilon_t$ indicates the white noise error term. While $\beta_1$ to $\beta_5$ and $\alpha_1$ is $\alpha_5$ represented the long- and short-term dynamic, respectively. We investigated the long-run relationship between the series after getting the F-statistic value using the ARDL bound testing equation. The following hypotheses for the model were used to determine the long-run relationship between variables:

H0: $\beta1 = \beta2 = \beta3 = \beta4 = \beta5 = 0$ (no cointegration)

H0: $\beta1 \neq \beta2 \neq \beta3 \neq \beta4 \neq \beta5 \neq 0$ (cointegration)

We ran the long-run and short-run dynamics if there was cointegration identified among the variables that are H0: β1, β2, β3, β4, β5, β6 $\neq$0.

The following equations specify the long-run and short-run models of the ARDL specification:

**Long run**

$$\text{LN}EMP_t = \beta_0 + \sum_{i=1}^{P} \beta_1 \text{LN}EMP_{t-i} + \sum_{i=1}^{P} \beta_2 \text{LN}TA_{t-i} + \sum_{i=1}^{P} \beta_3 \text{LN}MC_{t-i} \\ + \sum_{i=1}^{P} \beta_4 \text{LN}FD_{t-i} + \sum_{i=1}^{P} \beta_5 \text{LN}TR_{t-i} + \varepsilon_t \qquad (4)$$

**Short-run**

$$\Delta \text{LN}EMP_t = \alpha_0 + \sum_{i=1}^{P} \alpha_1 \Delta \text{LNEMP}_{t-i} + \sum_{i=1}^{P} \alpha_2 \Delta \text{LN}TA_{t-i} + \sum_{i=1}^{P} \alpha_3 \Delta \text{LN}MC_{t-i} \\ + \sum_{i=1}^{P} \alpha_4 \Delta \text{LN}FD_{t-i} + \sum_{i=1}^{P} \alpha_5 \Delta \text{LN}TR_{t-i} + \mu ECM_{t-1} + \varepsilon_t \qquad (5)$$

Where β and α is the long run and short run dynamics coefficient, respectively; while μ is the coefficient of the speed of adjustment and $\varepsilon_t$ is the disturbance term.

**The asymmetric analysis: NARDL.** This paper also employs the NARDL technique developed by Shin, Yu and Greenwood-Nimmo [67] to determine the probable asymmetric association among the variables ignored by the linear ARDL model. The NARDL model, like the ARDL model, has criteria for the integration order of the variables. Thus, the following Shin, Yu and Greenwood-Nimmo [67], Eq 3 can be restated in the following form:

$$\Delta \text{LNEMP}_t = \beta_0 + \beta_1 \text{LN}EMP_{t-1} + \beta_2^+ \text{LN}TA_{t-1} + \beta_3^- \text{LN}TA_{t-1} + \beta_4^+ \text{LN}MC_{t-1} + \beta_5^- \text{LN}MC_{t-1} +$$

$$\beta_6 \text{LN}FD_{t-1} + \beta_7 \text{LN}TR_{t-1} + \sum_{i=1}^{P} \theta_i \Delta \text{LNEMP}_{t-i} + \sum_{i=1}^{P} \theta_i^+ \Delta \text{LN}TA_{t-i} + \sum_{i=1}^{P} \theta_i^- \Delta \text{LN}TA_{t-i} +$$

$$\sum_{i=1}^{P} \theta_i^+ \Delta \text{LN}MC_{t-i} + \sum_{i=1}^{P} \theta_i^- \Delta \text{LN}MC_{t-i} + \sum_{i=1}^{P} \theta_i \Delta \text{LN}FD_{t-i} + \sum_{i=1}^{P} \theta_i \Delta \text{LN}TR_{t-i} +$$

$$\varepsilon_t \qquad (6)$$

From Eq 6, $\beta_i^+$, $\beta_i^-$ and $[\sum_{i=1}^{P} \theta_i^+]$, $[\sum_{i=1}^{P} \theta_i^-]$, captures the long- and short-run positive and negative impact of tourist arrival (TA) and market capital (MC) on service sector employment (EMP). Like the ARDL model, the bound test is restored to determine whether the variables are asymmetrically cointegrated or not. Furthermore, the Wald-test is used to assess the long (short-run) asymmetric linkage $\beta = \beta^+ = \beta^-$ ($\theta = \theta^+ = \theta^-$) for both tourist arrival and market capital. The short-run asymmetric association can be provided via the dynamic multiplier effect in the following method, given the validation of the non-linear relationship.

$$D_S^+ = \sum_{j=0}^{s} \frac{\omega \, LN \, EMP_{i-j}}{\omega \, LN \, TA_{t-i}^+},$$

$$D_S^- = \sum_{j=0}^{s} \frac{\omega \, LN \, EMP_{i-j}}{\omega \, LN \, TA_{t-i}^-},$$

s = 0, 1, 2, 3, . . . . . . . . . . . nothing that s → ∞, $D_S^+ = \beta_2^+$, $D_S^- = \beta_3^-$,

$$D_S^+ = \sum_{j=0}^{s} \frac{\omega \, LN \, EMP_{i-j}}{\omega \, LN \, MC_{t-i}^+},$$

$$D_S^- = \sum_{j=0}^{s} \frac{\omega \, LN \, EMP_{i-j}}{\omega \, LN \, MC_{t-i}^-},$$

s = 0, 1, 2, 3, . . . . . . . . . . . nothing that s → ∞, $D_S^+ = \beta_4^+$, $D_S^- = \beta_5^-$

## Results

The study represented the descriptive statistics and correlations of the variables, as shown in Table 2. According to Jarque–Bera statistics, the series of employment, tourism arrivals,

**Table 2. Descriptive analysis.**

|  | LNEMP | LNTA | LNMC | LNFD | LNTR |
|---|---|---|---|---|---|
| Mean | 4.317028 | 15.46061 | 27.16119 | 4.605408 | 3.693505 |
| Median | 4.320151 | 15.51351 | 27.37794 | 4.689457 | 3.714279 |
| Maximum | 4.361569 | 16.0632 | 28.04211 | 4.958803 | 3.824238 |
| Minimum | 4.26465 | 14.67861 | 25.62967 | 4.098294 | 3.473672 |
| Std. Dev. | 0.027998 | 0.348517 | 0.788039 | 0.281508 | 0.084236 |
| Skewness | -0.20182 | -0.30729 | -0.45122 | -0.43954 | -0.89907 |
| Kurtosis | 2.097516 | 2.862972 | 1.798964 | 1.7853 | 3.599484 |
| Jarque-Bera | 1.181036 | 0.479087 | 2.727051 | 2.716663 | 4.341166 |
| Probability | 0.55404 | 0.786987 | 0.255758 | 0.257089 | 0.114111 |
| Sum | 125.1938 | 448.3576 | 787.6746 | 133.5568 | 107.1117 |
| Sum Sq. Dev. | 0.021949 | 3.400997 | 17.38817 | 2.218901 | 0.198682 |
| Observations | 29 | 29 | 29 | 29 | 29 |

market capital, financial development, and trade are normally distributed. The low standard deviation values for all variables indicate that the data are spread around the mean rather than widely dispersed, validating the normal distribution results as determined by Jarque–Bera values.

## Unit root test

The study exhibited the ADF unit root test to determine the order of integration, whether the study's variables were stationary at the first difference I (1). The result of ADF in Table 3 shows that all the variables were stationary at first difference I (1). Hence, the series of variables is shown to have a valid long-run relationship.

## BDS test results

Table 4 illustrates the result of the BDS test for non-linearity. For all levels, P-value is found to be less than 0.05, thus the null hypothesis is rejected, implying that the series are linearly dependent. The results suggest that Australia's employment, tourism arrivals, market capital, financial development, and trade are non-linearly dependent.

## Bound testing results

The study employed the ARDL bound test to examine the association between variables. As shown in Table 5, the F-statistics in the bound test is larger than 5% critical value, showing

**Table 3. ADF test with structural break: Additive and innovative outliers.**

| Variable | Augmented Dickey-Fuller test statistic | | | |
|---|---|---|---|---|
|  | Level | Probability | 1st difference | Probability |
| LNEMP | -1.104 | 0.699 | -7.304 | 0.000*** |
| LNTA | -2.034 | 0.272 | -3.251997 | 0.028** |
| LNMC | -1.56965 | 0.484 | -5.70256 | 0.000*** |
| LNFD | -2.58472 | 0.108 | -2.65109 | 0.096* |
| LNTR | -2.55841 | 0.113 | -5.68376 | 0.000*** |

Note: ***1% level of significance, ** 5% level of significance and * 10% level of significance.

**Table 4. BDS test results.**

| Variable | m = 2 | m = 3 | m = 4 | m = 5 | m = 6 |
|----------|-------|-------|-------|-------|-------|
| LNEMP | 0.182*** | 0.316*** | 0.406*** | 0.464*** | 0.502*** |
| LNTA | 0.173*** | 0.287*** | 0.355*** | 0.402*** | 0.450*** |
| LNMC | 0.156*** | 0.276*** | 0.359*** | 0.424*** | 0.469*** |
| LNFD | 0.201*** | 0.338*** | 0.431*** | 0.496*** | 0.547*** |
| LNTR | 0.134*** | 0.206*** | 0.267*** | 0.297*** | 0.298*** |

Note

***1% level of significance.

that the null hypothesis of no level relationship is rejected at the significance level. This indicates a cointegration relationship between variables at least at 1% significance level. Therefore, the result allows us to apply the ARDL and NARDL cointegration approaches. The implementation of cointegration across variables allows us to examine the impact of tourist arrival, market capital, financial development and trade on service sector employment in the short and long run.

## Long-run results

In the long run ARDL model, outlined in Table 6, it is evident that that tourist arrivals affect service sector employment significantly and positively. With 1% growth in tourist arrivals, employment in the service sector increases by 0.037%. As for an asymmetric TA-EMP linkage, the NARDL model shows that a positive shock in TA (LNTA+) increases employment in the service sector while a negative shock TA (LNTA-) reduces employment. Market capital significantly and negatively affects service sector employment in a symmetric linear model (ARDL); however, in the asymmetric model (NARDL), positive shock in MC (LNMC$^+$) can lead to an increase in employability in the long run. In the long run, the symmetric ARDL model both financial development and trade, show a significant positive impact on service sector employment while no significant and negative association is found for financial development and trade, respectively, in the NARDL model.

## Short-run results

The short-run linear (ARDL) and non-linear (NARDL) dynamics are presented in Table 7. In the short run, linear (ARDL) dynamics revealed that tourist arrival, market capital, and trade volume exposed a statistically significant positive coefficient of 0.095, 0.005 and 0.041. However, financial development showed a negative coefficient -0.013. As for asymmetric EMP-tourist arrival linkage, while a positive shock in TA ($\Delta$LNTA$^+$) increases employment, a negative shock in tourist arrival decreases employment in the service sector. A positive shock in

**Table 5. Bounds test for the nonlinear cointegration.**

| Series | F-statistics | LCB I (0) | UCB I (1) | Conclusion |
|--------|--------------|-----------|-----------|------------|
| ARDL: LNEMP = f (LNTA, lnMC, lnFD, lnTR) | 23.950*** | 3.29 | 4.37 | Cointegrated |
| NARDL: LNEMP = f (LNTA$^+$, LNTA$^-$, lnMC$^+$, lnMC$^-$ lnFD, lnTR) | 19.016*** | 2.88 | 3.99 | Cointegrated |

Note

***1% level of significance, LCB = Lower Critical Bound, and UCB = Upper Critical Bound

**Table 6. Long run result of ARDL and NARDL.**

| Variable | ARDL (linear) Results | | NARDL (non-linear) Results | |
|---|---|---|---|---|
| | Coefficient | T-statistics | Coefficient | T-statistics |
| C | | | 4.295*** | 96.124 |
| LNTA | 0.037*** | 172.268 | - | - |
| LNTA$^+$ | - | - | 0.057*** | 24.483 |
| LNTA$^-$ | - | - | -0.083** | -3.337 |
| LNMC | -0.019*** | -388.601 | - | - |
| LNMC$^+$ | - | - | 0.005** | 2.761 |
| LNMC$^-$ | - | - | 0.006 | 1.953 |
| LNFD | 0.098*** | 578.139 | 0.011 | 1.924 |
| LNTR | 0.028** | 24.229 | -0.024* | -2.574 |

Note

***1% level of significance

** 5% level of significance and

* 10% level of significance.

MC can increase service sector employment in the short run; however, no significant association is found if there is any negative shock.

## Diagnostic tests

As shown in Table 8, diagnostic tests were used to assess the estimates' reliability. The diagnostic tests using the log transformation of time-series data are shown in the table. The Breusch-Godfrey Lagrange multiplier test found no serial connection, indicating unrelated observations. The Breusch-Pagan-Godfrey heteroscedasticity test demonstrated that the observation had no regression errors. The series was shown to be normally distributed by the Jarque-Bera normality test, and finally, the Ramsey RESET stability test confirmed the model as correctly specified.

**Table 7. Short-run results of ARDL and NARDL.**

| Variable | ARDL (linear) Results | | NARDL (non-linear) Results | |
|---|---|---|---|---|
| | Coefficient | T-statistics | Coefficient | T-statistics |
| C | 5.449*** | 176.639 | 12.579** | 5.415 |
| ΔLNTA | 0.095** | 133.997 | - | - |
| ΔLNTA$^+$ | - | - | 0.168** | 5.153 |
| ΔLNTA$^-$ | - | - | -0.243** | -2.800 |
| ΔLNMC | 0.005** | 69.468 | - | - |
| ΔLNMC$^+$ | - | - | 0.015** | 3.077 |
| ΔLNMC$^-$ | - | - | 0.018 | 2.061 |
| ΔLNFD | -0.013** | -19.683 | 0.033 | 1.629 |
| ΔLNTR | 0.041** | 52.785 | -0.170*** | -8.294 |

Note

***1% level of significance

** 5% level of significance and

* 10% level of significance.

**Table 8. Diagnostic test.**

| Variable | P-value | Result |
|---|---|---|
| **Breusch-Godfrey Serial Correlation LM Test** | 0.3694 | No serial correlation |
| **Breusch-Pagan-Godfrey (Hetroscedasticity Test)** | 0.8576 | No evidence of heteroskedasticity |
| **Jarque-Bera (Normality Test)** | 0.343 | Residuals are normally distributed |
| **Ramsey RESET Test (Stability Test)** | 0.6977 | Model is correctly specified |

### Stability test

The CUSUM and CUSUM of squares tests on the recursive residuals are used in this paper to assess the consistency of short-run beta coefficients in the ARDL method. The CUSUM test identifies orderly fluctuations in regression coefficients, whereas the CUSUM squares test identifies rapid fluctuations in regression coefficients that may change their stability. Figs 2 and 3 display the CUSUM and CUSUM square test results, indicating that all values were within the critical boundaries at a 5% significant level.

### Dynamic multiplier graph

A dynamic multiplier graph for NARDL is presented in Figs 4 and 5 to analyse the adjustment of asymmetry in the existing long-run equilibrium after passing to a new long-run equilibrium due to negative and positive shocks. The asymmetry curves depict a linear mixture of dynamic multipliers resulting from positive and negative shocks of tourist arrival and market capital.

### Discussion

The present study aimed to investigate the symmetric and asymmetric relationship of service sector employment with tourist arrival, market capital, financial development, and international trade in Australia. To check the stationarity of the variables, the ADF unit root was implemented. In addition, the ARDL bound testing approach was used to check the symmetric dynamic linkage among the variables, while NARDL bound testing was implemented to investigate the asymmetric dynamic linkages. Tourist arrival, financial development and trade were found to be positively cointegrated; however, market capital showed an inverse association

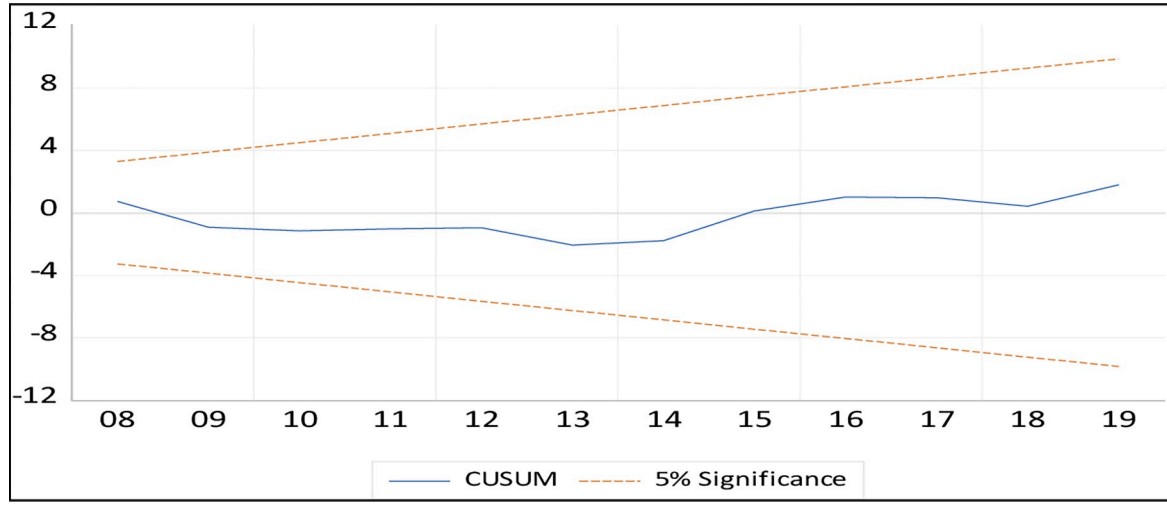

**Fig 2. CUSUM test.**

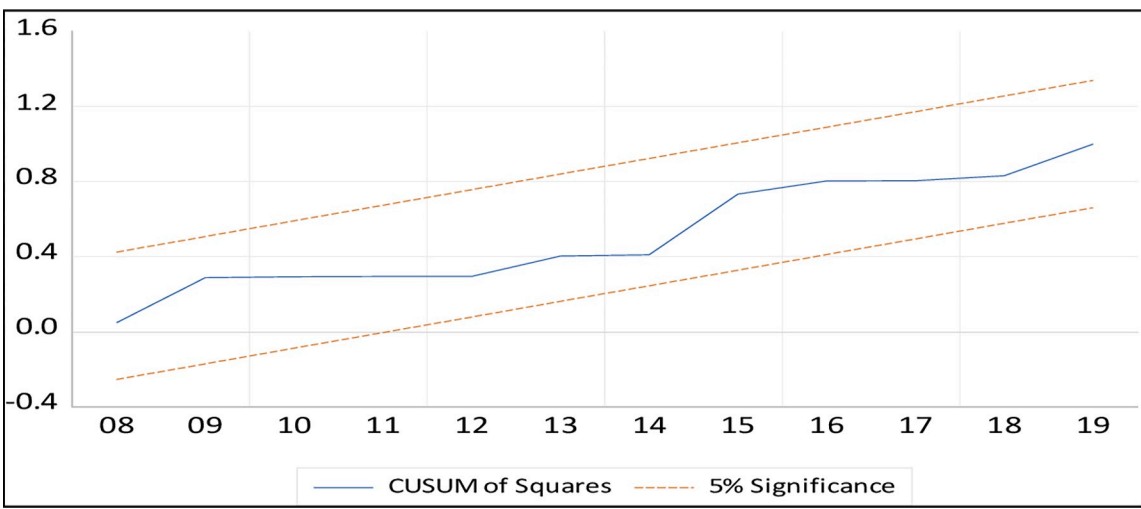

**Fig 3. CUSUMQ test.**

with service sector employment in the long run ARDL dynamics. A long run asymmetric NARDL relationship reaffirmed the service sector employment and tourist arrival linkage, while a positive shock in TA increased employment, and a negative shock decreased employment in the service sector. In addition, in the long run NARDL dynamics, trade was found to be negatively cointegrated with employability in the service sector.

This study finds a significant positive short and long-run effect between tourist arrival and service sector employment in Australia. This finding is consistent with similar studies conducted in Pakistan [70], India [71], and Africa [5], in all of which it has been observed that international tourism directly impacts the economy and therefore employment. Tourism is a critical component of the service industry since it contributes to the development of hotels, restaurants, transportation, and other associated services [70]. Thus, tourism contributes to

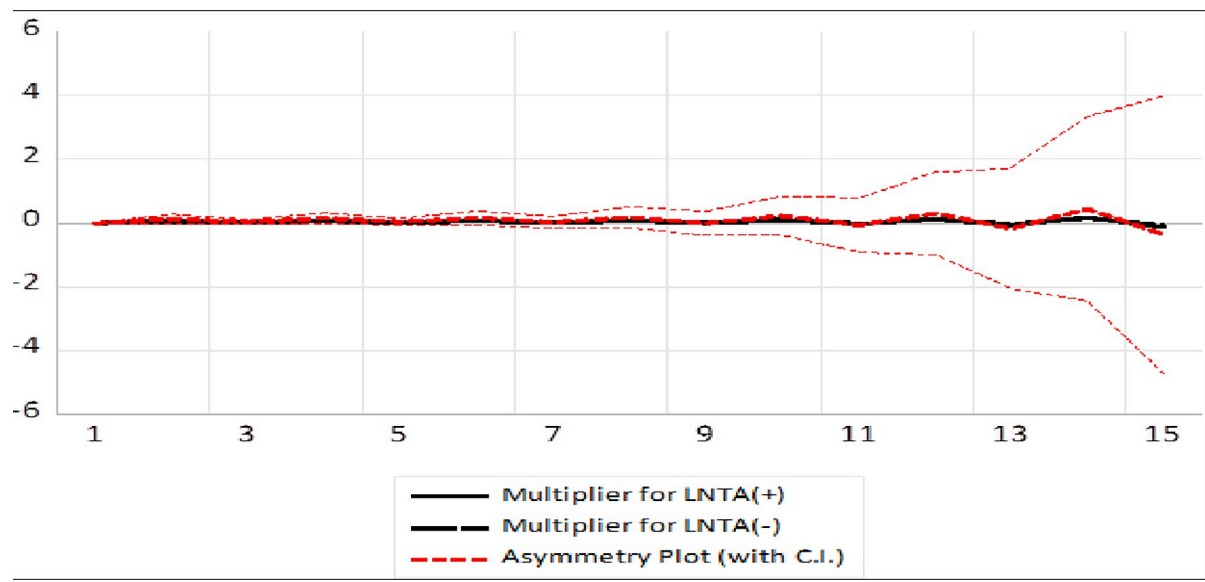

**Fig 4. NRDL dynamic multiplier graph for tourist arrivals.**

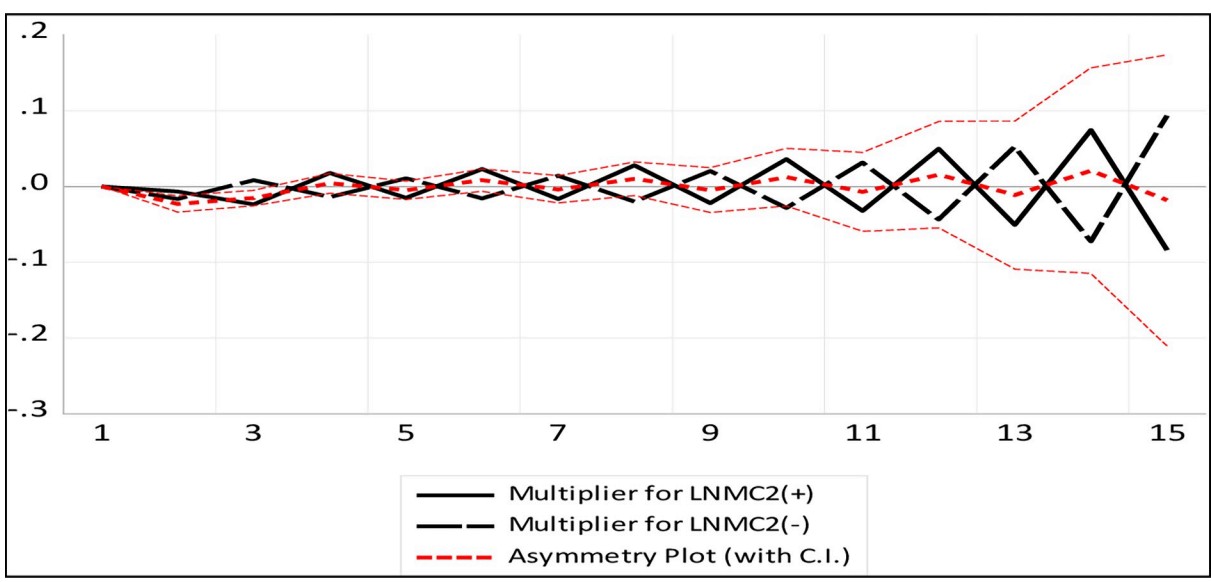

**Fig 5. NRDL dynamic multiplier graph for market capital.**

the economy by creating jobs and boosting GDP [71]. Tourism has had a large indirect impact on economic development, contributing to the market, improving human living standards, raising government revenue through income and taxation, and even expanding the production of goods and services in Hong Kong, in addition to its direct benefits [72].

Furthermore, this study explored the relationship between market capital and employability in the service sector and found a negative association in ARDL long-run dynamics. Notably, no existing studies have previously used Australian data; therefore, this is the first study to conclude that tourism affects service sector employment in Australia. However, a study conducted on nine European countries found that a fall in capital stock increased overall unemployment [73]. In some other studies, the relationship between the stock market and economic growth has been analysed. For example, a structured stock market has been found to result in a country's economic growth in the long run, and stock market-based finance is a determinant of economic growth [26, 27]. The possible explanation of the negative relationship between market capital and service sector employment in Australia might be due to the difference in volume of stocks in manufacturing and service-oriented firms. The presence of banking and non-banking financial institutions and telecoms are dominant in various stock exchanges, while the contribution of other service-oriented industries is insignificant. Hence, the rise in market capital is more clearly linked with the entrance of the manufacturing industry in the stock market; thus, employability increase in this sector is more likely.

Financial development was found to affect service sector employment positively in Australia. Although earlier studies in Australia explored the dynamic, positive relationship between financial development and economic growth [45, 74], research focused on financial development impact on service sector employment was limited. It is expected that since financial development affects economic growth, the service sector, which is a part of GDP, will also increase.

The findings of this study, concerning trade, is aligned with the findings of Jansen and Lee [46] and Vandenberg [47] in which a bidirectional impact of trade on employment was found. However, employment in the service sector requires explicitly further investigation. Concerning service sector employment, the case of Australia is somehow similar to the case of China as

found by Yu and Meng [50]. Inter-country trade, as found in the case of Germany, does not influence the rise of service sector employment [52], which may remain true for Australia; however, it also requires further investigation.

## Conclusion and policy implications

This study examined the symmetric and asymmetric relationships of tourist arrival, market capital, financial development and international trade with Australian service sector employment using annual data from 1991 to 2019. Unlike prior research on Australia, this was among the primary attempts to separately examine the impact of the mentioned variables on service sector employment separately. The linear ARDL results imply that tourist arrival, financial development and trade has a long-term positive effect on service sector employment, whereas market capital negatively affects service sector employment in Australia. However, the NARDL results show that a rise in tourist arrival intensifies service sector employment in the long run, whereas a fall shrinks employability. Further investigations may be conducted to assess the impact of inter-country trade, especially with the countries from Asia and the Pacific countries.

Based on the findings, a few policy recommendations are suggested. The government should plan and initiate tools to ensure consistent tourist arrivals throughout the year to keep service sector employment predictable. Reduction of visa restrictions and/or easing entry requirements may be some initiatives to promote tourist arrivals. In addition, credit towards the service industry, especially in tourism sector, should be supported since domestic credit to private sectors results in higher service sector employment [35]. The national government should also promote service exports to other countries to enjoy a higher level of employment in overall service sector. Proper attention should be given for financial development as it can positively contribute to employment level. The federal government should also ensure the dignity of the indigenous people. For all these measures, both short and long-term plans should be undertaken and executed carefully in Australia.

## Acknowledgments

This paper uses the data from World Development Indicators from World Bank. The authors would like to thank World Bank for providing the data. The authors also would like to thank Dr Barbara Harmes for proofreading the manuscript before submission.

## Author Contributions

**Conceptualization:** Avishek Khanal, Mohammad Mafizur Rahman, Rasheda Khanam, Eswaran Velayutham.

**Data curation:** Avishek Khanal, Eswaran Velayutham.

**Formal analysis:** Avishek Khanal, Mohammad Mafizur Rahman, Eswaran Velayutham.

**Investigation:** Avishek Khanal.

**Methodology:** Avishek Khanal, Eswaran Velayutham.

**Project administration:** Avishek Khanal.

**Resources:** Avishek Khanal.

**Software:** Avishek Khanal.

**Supervision:** Mohammad Mafizur Rahman, Rasheda Khanam, Eswaran Velayutham.

**Validation:** Avishek Khanal.

**Visualization:** Avishek Khanal.

**Writing – original draft:** Avishek Khanal.

**Writing – review & editing:** Avishek Khanal, Mohammad Mafizur Rahman, Rasheda Khanam, Eswaran Velayutham.

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
