## [Decision Letter · Decision Letter 0]

5 May 2022

PONE-D-22-05444The role of tourism in service sector employment: Do market capital, financial development and trade also play a role?PLOS ONE

Dear Dr. Avishek Khanal,

Thank you for submitting your manuscript to PLOS ONE. After careful consideration, we feel that it has merit but does not fully meet PLOS ONE’s publication criteria as it currently stands. Therefore, we invite you to submit a revised version of the manuscript that addresses the points raised during the review process.

We look forward to receiving your revised manuscript.

Kind regards,

Ricky Chee Jiun Chia

Academic Editor

PLOS ONE

Journal Requirements:

"NO"

"No" 

Reviewers' comments:

Reviewer's Responses to Questions

5. Review Comments to the Author

Reviewer #1: The study titled "The role of tourism in service sector employment: Do market capital, financial development and trade also play a role?" has been prepared in accordance with scientific rules. Thank you for giving me the opportunity to read and evaluate this work.

Reviewer #2: The study examined a very interesting subject using annual data from 1991 to 2019.

This topic is relevant from societal point of view.

The choice of statistical methods used in the study (ARDL, NARDL) is unique.

A full revision of the manuscipt is highly recommended for the following reasons:

1. The “2. Background and Literature Review” section is advised to be extended.

2. The topic considered (the role of tourism in service sector employment) has a large body of literature, and a thorough presentation of this literature would make the article more valuable. If the authors do not wish to extend the range of sources discussed, it is pertinent to define and justify those used more precisely.

3. Policy recommendations were suggested in section 6. (Conclusion and policy implications). I think these are not sufficiently substantiated and reasoned for.

4. Your proposals should be more specific.

5. A thorough proofreading is strongly recommended regarding grammatical errors and style.

---

## [Author Response · Author response to Decision Letter 0]

16 Jun 2022

Response to Reviewers' Comments

1st June 2022

Editor-in-Chief

Re: Responses /Actions to the Reviewers’ Report on Ms. Ref. No.: PONE-D-22-05444

Title: The role of tourism in service sector employment: Do market capital, financial development and trade also play a role?

Dear Editor,

We are grateful to the respected editor for giving us an opportunity to revise and resubmit the paper. We think the comments are useful for further development of the paper. We have tried our level best to address all the comments of the reviewers in our revised manuscript. We believe our joint effort will enrich the paper in terms of quality. Thanks a lot, to the editor and the reviewers for your support, cooperation and valuable suggestions in the work. The Comments/Suggestions and our Responses/Actions are noted below for your consideration.

Reviewer #1: The study titled "The role of tourism in service sector employment: Do market capital, financial development and trade also play a role?" has been prepared in accordance with scientific rules. Thank you for giving me the opportunity to read and evaluate this work.

Response: Thank you very much for your nice comment and interest in reading our manuscript. 

Reviewer #2: The study examined a very interesting subject using annual data from 1991 to 2019.

Response: Thank you very much for showing your interest in reading our manuscript. 

A full revision of the manuscipt is highly recommended for the following reasons:

1. The “2. Background and Literature Review” section is advised to be extended.

Response: We have reviewed some recent literature to improve the section (See page# 4-8). Followings are the literatures we have added: 

1. Camargo, B. A., Winchenbach, A., & Vázquez-Maguirre, M. (2022). Restoring the dignity of indigenous people: Perspectives on tourism employment. Tourism Management Perspectives, 41, 100946.

2. Ioannides, D., Gyimóthy, S., & James, L. (2021). From liminal labor to decent work: A human-centered perspective on sustainable tourism employment. Sustainability, 13(2), 851.

3. Tosun, C., Çalişkan, C., Şahin, S. Z., & Dedeoğlu, B. B. (2021). A critical perspective on tourism employment. Current Issues in Tourism, 1-21.

4. Walmsley, A., Koens, K., & Milano, C. (2021). Overtourism and employment outcomes for the tourism worker: impacts to labour markets. Tourism Review.

5. Sayan, S., & Alkan, A. (2021). A novel approach for measurement and decomposition of the economywide costs of shutting down tourism and related service sectors against COVID-19. Tourism Economics, 13548166211037100.

6. Scarlett, H. G. (2021). Tourism recovery and the economic impact: A panel assessment. Research in Globalization, 3, 100044.

7. Yousaf, U. B., Ullah, I., Wang, M., Junyan, L., & Rehman, A. U. (2021). Does board capital increase firm performance in the Chinese tourism industry?. Corporate Governance: The International Journal of Business in Society. DOI 10.1108/CG-04-2021-0165 

8. Dalwai, T., & Sewpersadh, N. S. (2021). Intellectual capital and institutional governance as capital structure determinants in the tourism sector. Journal of Intellectual Capital.

9. Isaeva, A., Salahodjaev, R., Khachaturov, A., & Tosheva, S. (2022). The impact of tourism and financial development on energy consumption and carbon dioxide emission: Evidence from post-communist countries. Journal of the Knowledge Economy, 13(1), 773-786.

10. Ehigiamusoe, K. U. (2021). The nexus between tourism, financial development, and economic growth: Evidence from African countries. African Development Review, 33(2), 382-396.

11. Irani, F., Katircioglu, S., & Gokmenoglu, K. K. (2021). Effects of business and finance conditions on tourism firms’ financial performances: evidence from major tourist destinations. SAGE Open, 11(3), 21582440211040120.

2. The topic considered (the role of tourism in service sector employment) has a large body of literature, and a thorough presentation of this literature would make the article more valuable. If the authors do not wish to extend the range of sources discussed, it is pertinent to define and justify those used more precisely.

Response: While addressing the section: Background and Literature Review, this suggestion has automatically been incorporated (See page# 4-8); Thanks to the reviewer for the suggestion. 

3. Policy recommendations were suggested in section 6. (Conclusion and policy implications). I think these are not sufficiently substantiated and reasoned for.

Response: We have tried to link the suggestions with the findings of the research (See page# 19 last paragraph). 

4. Your proposals should be more specific.

Response: We have included specific suggestions based on the findings of our research (See page# 19 last paragraph). 

5. A thorough proofreading is strongly recommended regarding grammatical errors and style.

Response: Thanks reviewer for this suggestion. Language has been checked by native speaker and proof-reader.

---

## [Editor Report · Decision Letter 1]

17 Jun 2022

The role of tourism in service sector employment: Do market capital, financial development and trade also play a role?

PONE-D-22-05444R1

Dear Dr. Avishek Khanal,

We’re pleased to inform you that your manuscript has been judged scientifically suitable for publication and will be formally accepted for publication once it meets all outstanding technical requirements.

Kind regards,

Ricky Chee Jiun Chia

Academic Editor

PLOS ONE
---

## [Editor Report · Acceptance letter]

28 Jul 2022

PONE-D-22-05444R1 

The role of tourism in service sector employment: Do market capital, financial development and trade also play a role? 

Dear Dr. Khanal:

I'm pleased to inform you that your manuscript has been deemed suitable for publication in PLOS ONE. Congratulations! Your manuscript is now with our production department. 

Kind regards, 

on behalf of

Dr. Ricky Chee Jiun Chia 

Academic Editor

PLOS ONE